# Deconstruction of the Green Bubble during COVID-19 International Evidence

Bikramaditya Ghosh [1] , Spyros Papathanasiou [2],*, Vandita Dar [1] and Dimitrios Kenourgios [2]

1   Symbiosis Institute of Business Management, Symbiosis International (Deemed University) Electronic City, Hosur Road, Bengaluru 560100, Karnataka, India; b.ghosh@sibm.edu.in (B.G.); vandita.dar@sibm.edu.in (V.D.)
2   Department of Economics, School of Economics and Political Sciences, National and Kapodistrian University of Athens, 1 Sofokleous Street, 10559 Athens, Greece; dkenourg@econ.uoa.gr
*   Correspondence: spapathan@econ.uoa.gr

**Abstract:** Bubbles are usually chaotic but can be predictable, provided their formation matches the log periodic power law (LPPL) with unique stylized facts. We investigated Green Bubble behaviour in the stock prices of a selection of stocks during the COVID-19 pandemic, namely, those with the highest market capitalization from a basket of North American and European green energy or clean tech companies and the S&P Global Clean Energy Index. Moreover, the biggest Exchange Traded Fund (TAN) by market capitalization was also considered. The examined period is from 31 December 2019 to 11 October 2021, during which we detected 35 Green Bubbles. All of these followed the LPPL signature while calibrated through the 2013 reformulated LPPL model. In addition, the average drawdown emerged as four times that of the regular S&P-500 stock index (108% vs. 27%) under stressed conditions, such as the COVID-19 pandemic (stylized fact). Finally, the aftermaths of Green Bubbles, unlike regular bubbles, are not destructive, as these bubbles increase economic activity and infrastructure spending and are hence beneficial for holistic growth (described as *Social Bubble Hypothesis*). We document that there are benefits in adapting greener and more sustainable business models in energy production. Green and sustainable finance offers benefits and opportunities for stock exchanges, especially for energy stocks. As a result, many businesses are focusing on sustainability and adopting an eco-friendly business model, which helps the environment, helps sustainability and attracts investors.

**Keywords:** social bubble hypothesis; sustainable development; log periodic power law; green finance; green energy; COVID-19

## 1. Introduction

Green and sustainable development and finance offer benefits and opportunities for stock exchanges. Green finance products have already been introduced in many markets and have seen extraordinary growth. "Green finance" is a broad term which refers to the flow of financial investments into sustainable development projects and initiatives, environmental products, and policies that encourage the development of a more sustainable economy. "Green finance" includes—but is not limited to—climate finance [1].

Subsequently, green stocks and green bonds are outperforming their non-green benchmarks. As a result, many firms are focusing on sustainability and adopting an eco-friendly business model, which helps the environment, helps sustainability and attracts investors. This increase is partly attributed to a growing consensus about the role of stock exchanges in promoting economic development.

Green or renewable energy is the future of our Earth. It reduces carbon emissions and global warming by generating electricity or heat from renewable or low-carbon sources using any equipment whose capacity to generate electricity or heat does not exceed the capacity specified (in relation to the production of electricity, 5 megawatts) [2]. Most

corporations in the West have either partially or completely transformed their operations following these mandates. We indicate two things under the mandates: 1. Following Sustainable Development Goals (SDGs) and 2. Trading carbon rights through Cap and Trade in order to reduce the carbon footprint.

Prima facie, all agree on the necessity of green finance; however, it requires large structural changes and thus invites a fair share of financial instability; in fact it is completely different for sunrise industries vis-a-vis sunset industries [3] (Semieniuk et al., 2021). Moreover, the abetment cost of transitioning into a state where fossil-fuel-based energy is completely avoided, due either to technological reasons or a standpoint of behavioural acceptance, would be rather high. Interestingly, should green energy be more affordable in the future, it would result in an increase in the dynamic cost of the green energy infrastructure [4] (Gillingham & Stock, 2018).

However, it has been reported that green energy is no longer a niche. Some of the largest conglomerates on Earth plan to invest in green energy in a big way. Global investment has been shifting away from fossil fuels at a rapid speed during the COVID-19 outbreak. Investment funds focusing on the environment have seen a USD 2 trillion shift globally in the first quarter of 2021 alone. This is more than a three-fold growth over a few years. A little over USD 5 billion worth of bonds are designed to fund green technology companies daily [5]. Therefore, it has become imperative to control the plausible formation of bubbles when so much money is drawn to the sector. Moreover, their evaluation rose as well. Green Energy companies witnessed an astronomical growth in their revenue throughout 2020. In fact, it almost doubled from $6.7\times$ in Q1 to $12.7\times$ in Q4 before stabilizing around the $10\times$ mark [6]. Some of the largest investment bank corporations, such as BlackRock, have commented that the investment market is most probably experiencing a "tectonic shift" towards green alternatives. To add to the same debate, global investment funds are chasing after environmental, social and governance principles (ESG) [7] (Umar, et al., 2020). In 2020 alone, these attracted USD 350 bn, compared to USD 165 bn in 2019 [8].

We need to look intuitively into this matter. With the constant flow of positive feedback and reaffirmation of the very fact that the future of energy is "green", it has become evident that we are witnesses to a speculative bubble formation. Furthermore, during COVID-19, positive feedback loops have ensured an enormous amplification of capital flight to green energy companies in a very short amount of time, with an unlikely boost coming from news agencies. Herding and the pandemic have quite a few things in common; in fact, they share a similar logistic curve pattern. Decisions which are perfectly rational at an individual capacity remain hardly rational in a group situation [9] (Shiller, 2000).

The cardinal idea behind this work is to find out whether the theory of the Green Bubble exists in reality or not. We have taken a pluralistic approach by accommodating contradictory literature, basing, however, our conclusion on empirical evidence. Many studies [10,11] (Dempse & Edwards, 2008; Belle, 2007) have suggested that too much money is chasing too few green technology stocks, thus forming an uncanny bubble. Researchers have coined the term for this as a "Green Bubble" [10–20] (Belle, 2007; Bennett, 2010; Dempsey & Edwards, 2008; Koutsokostas & Papathanasiou, 2017; Koutsokostas, et al., 2019; Christopoulos, et al., 2014; Christopoulos, et al., 2019; Koutsokostas, et al., 2018; Wimmer, 2016). However, inconsistent observations over a rather short period of time cannot even create a "stylized fact", let alone a "theory". This study intends to unearth the feasibility behind such claims.

Giorgis et al. (2021) [21] coined the term "Clean Tech Bubble" instead of "Green Bubble", albeit the meaning is the same.

"Cleantech" or "Clean Tech" is generally defined as the products or services based on knowledge that improves operational performance, productivity or efficiency, while at the same time reducing costs, inputs, energy consumption, waste or pollution [22,23].

Green assets are on a significant rise. The prices of metals for battery production such as lithium and cobalt have risen by approximately two-thirds and one-third, respectively.

This sudden increase in prices during the pandemic forms a stretched valuation, which is indicative of a "Green Bubble" [12].

According to Giorgis et al. (2021) [21], despite the fact that all types of speculative bubbles are apparently disruptive in nature, some have lasting positive impacts. It is a well-documented fact that most bubbles germinate from irrational exuberance [9,24] (Shiller, 2000; Vasiliou, et al., 2008). Social bubbles accelerate capital spending, infrastructure building and enhance usage of emerging technologies [25,26] (Garber, 2001; Kindelberger & Aliber, 2005). It springs from the innumerable positive interactions in influential social circles (select government agencies, venture capitalists and high-net-worth individuals), thus reinforcing positive feedback with a significant impact. In this way, it reaches widespread acceptance and extraordinary commitments [27] (Gisler, et al., 2011). Hence, social bubbles are far from the destruction of public wealth in the long run.

This study contributes to the existing literature, as it is the first attempt to search for a common thread across the financial bubbles of the stock prices of a selection of stocks from a basket of Green Energy or Clean Tech companies (from North America and Europe) and the S&P Global Clean Energy Index during COVID-19 by using the Filimonov and Sornette (2013) [28] modified LPPL. In addition, we differentiated from previous studies [29–32] (Geuder et al., 2019; Ghosh et al., 2021; Ghosh, et al., 2020; Wheatley et al., 2019) by testing the robustness of the LPPL following the reformulated version of the LPPL calibrations proposed by Filimonov and Sornette (2013) [28].

"Clean energy" is defined as energy generated from Solar, Hydro, Bio, Geothermal, Wind, Nuclear and Hydrogen sources. It mostly overlaps with the term "Green energy".

Global stock markets are fractal, as proved by Edger Peters following Mandelbrot's principle [33] (Mandelbrot, 1963). The LPPL is a micro-state investigation of a fractal system, exhibiting macro-state phenomena. Therefore, the LPPL works well for most global asset classes that are stochastic in nature and follow a log normal distribution, including the stocks of the green energy basket under consideration [21,33–35] (Peters, 1994; Watkins & Franzke, 2017).

Therefore, the paper is composed of five sections. Section 2 briefly presents the literature review. Section 3 describes the data and the methodology used. In Section 4, the outcomes and findings of the research are presented. Finally, in Section 5, the conclusions are reported.

## 2. Literature Review

The term "Bubble" has essentially been a well-researched topic, unlike the term "Green Bubble". Literature regarding the "Green Bubble" or the "Clean Tech Bubble" is growing post-COVID-19, albeit slowly. O'Hara (2008) [36] provided extremely interesting and diverse perspectives on what the term "bubble" actually meant as interpreted by experts over a period of time. O'Hara (2008) [36] delved into the explanations offered by Garber (2001) [25] about seven years earlier, who defined the bubble as "a fuzzy word filled with import but lacking any solid operational definition", implying that a bubble involved a movement in prices, which could not be adequately explained by fundamentals. While Garber (2001) [25] upheld the view that bubbles could be both positive as well as negative, that was generally not the popular opinion. Palgrave's *Dictionary of Political Economy (1926, p. 181)*, viewed a "bubble" as "any unsound undertaking accompanied by a high degree of speculation". Kindelberger & Aliber, 2005 [26] also seemed to be disparaging of bubbles when they stated that "a bubble is an upward price movement over an extended range that then implodes". Towing the same line, Brunnermeier (2009) [37] opined that "bubbles are typically associated with dramatic asset price increases, followed by a collapse".

In an earlier work, Garber (1990) [38] emphasized that one should always intensively look for reasonable economic explanations before classifying a speculative event as an inexplicable bubble. For instance, probable expectations of high returns could be fundamental and reasonable explanations stemming from sound economic analysis. Even when uninitiated market participants act upon price movements caused by insider trading, the

resultant movement in asset prices should be considered as fundamental and not a bubble, despite the fact that these perceptions could eventually be disproved. Using this reasoning, Garber (2001) [25] concluded that the Dutch Tulip Mania (1634–37), often referred to as a classic bubble, reflected normal pricing behaviour, especially since the tulip bulbs involved were rare, displaying unique patterns created by a mosaic virus. In fact, a more accurate version of the Tulip Mania unfolded in the beginning of 1637, when there was a steep and rapid price rise, followed by a collapse in the market for common bulbs dominated by the lower classes [25] (Garber, 2001). However, there was no evidence pointing out to any economic distress arising from this fact.

Some interesting work has been carried out of late, where Quinn and Turner intriguingly speak of a "bubble triangle", specifically highlighting the positive social impacts of some bubbles [39] (Quinn & Turner, 2020). The positive bubble has the potential to encourage and nurture innovation, leading to many more people becoming entrepreneurs, which may result in a virtuous growth cycle. Bubble companies may even bring in innovations and technological breakthroughs, which, going ahead, may be beneficial to many other industries as well. Bubbles by nature seem to create an environment that apparently attracts capital for technologically intensive projects, which can have tremendous positive economic impacts. Gisler et al. (2011) [27] illustrated the social bubble hypothesis in action in the Human Genome Project (HGP) right from the 1980s well into the 2000s, which led to the committed involvement of public as well as private entities, much beyond what was deemed possible through a standard rational cost–benefit analysis [27] (Gisler et al., 2011). The initiation of the HGP as a high-stakes public project created the necessary hype and aroused the interest of private players, who then joined the bandwagon. The social interactions of public and private entities created the right mix of collaboration and competition, thus creating a positive network of reinforcing feedbacks. The supporting analysis of the biotech sector in the financial stock markets endorsed the existence of bubbles; however, tangible outcomes from the hype and the aura surrounding these bubbles might not be reaped in the short to medium term. Bubbles may take a long time to actually yield substantial positive outcomes. Bubbles are repetitive and change phases past the critical point (crash). The valuation reaches a decent height through this repetitive process. However, events such as COVID-19 (an extreme event) typically increase their pace, thus reducing the chances of fully exploring their context.

Garber (2001) [25] also highlighted the fact that, many times, understanding bubbles becomes challenging because they lie at the intersection of finance, economics and psychology. The inexplicable in bubbles is thus very often attributed to market psychology or market sentiment, which is difficult to measure, yet it is a convenient explanation of phenomena that cannot be explained through fundamentals [25] (Garber, 2001). Aghion et al. (2009) [40] emphasized that given the enormity of climate change's challenges, the world required a gigantic push to clean-technology-related innovations and had woefully fallen short on this front, especially considering the EU-27 countries. This required a green public intervention along with private initiatives and a public policy that incentivized private green innovation. Needless to say, tackling the gargantuan climate adversity probably requires the creation of a bubble. While investment in innovative, breakthrough clean technology is the need of the hour, Knuth (2018) [41] stressed on the importance of finance being deployed to the right causes efficiently, the context being the billions of funds committed by Bill Gates and several other billionaires towards the development of breakthrough clean energy technologies. This Breakthrough Energy Coalition created in the run-up to the Paris COP21 climate summit in 2015 seemed basically flawed in ignoring the development of existing clean energy technology, such as solar and wind energy, while pushing for innovations in other clean technologies.

The debate, particularly in the United States (US), is about the clean energy transition strategy and whether it can largely be accomplished through a higher level of financialization for existing technologies, which would reduce the cost of capital and augment resource flow to the sector; otherwise a large-scale technological disruption is essential to make this

possible [41] (Knuth, 2018). The role of venture capital in supporting transformative clean energy technologies towards a sustainable future is explored by Marcus and his research team. Their evidence showed that venture capitalists were inclined to make bigger bets for longer periods on high-risk, clean energy technology firms [42] (Marcus et al., 2013). However, a noteworthy observation here was the tendency of venture capitalists to avoid conventional high-risk production, distribution, and installation companies, while focusing on those opportunities that were technologically intensive. These preliminary findings were also endorsed by Knuth (2018) [41], who highlighted the classic debate between conventional clean energy technologies versus highly innovative technologies, capable of creating breakthroughs and paradigm shifts.

An in-depth Australian study seemed to assume prophetic undertones when it highlighted an emerging sixth paradigm in the form of a speculative financial bubble, specifically in the renewable energy sector [43] (Mathews, 2013). Mathews continued his study, predicting that this specific bubble would lose steam sometime between 2015 and 2020. This would lead to an era of prudent and sustainable development of renewables by productive rather than financial capital. While the fourth paradigm was based on fossil fuels and centralized power generation, the fifth ushered a wave where Internet technologies were applied to the electric power grid.

Despite an increase in scholarly interest in the said subject, most studies [42,44–47] (Criscuolo & Menon, 2015; Zhong & Bazilian, 2018; Bürer & Wüstenhagen, 2008; Mrkajic et al., 2019; Marcus, et al., 2013) were found to focus on either venture capitalist funding or the overall impact of bubbles in various sectors. Neither Green Bubbles nor Clean Tech bubbles have been deconstructed to identify their embedded pattern, nor have they been evaluated through a new perspective (read as "Social Bubble Hypothesis") [21] (Giorgis, Huber, & Sornette, 2021). Fundamentally, bubbles are born on a spiral of positive feedbacks (sometimes irrational) and enjoy negative connotations with specific exceptions, such as the "Green Bubble" (through the lens of the Social Bubble Hypothesis). Moreover, bubble prediction relies on the accurate calculation of an optimum number of observations (5 days to 750 days) which need to be taken into consideration as per Filimonov & Sornette (2013) [28]. The explicit identification of past bubbles with greater accuracy would hold the key to building rational "stylized facts" for future use. We have conducted all these things in our following segments.

## 3. Data and Research Methodology

Past studies [48–54] (Bree & Joseph, 2013; Johansen et al., 2000; Johansen and Sornette, 2010; Johansen and Sornette, 2001; Kenourgios et al., 2021; Samitas et al., 2022; Sornette and Johansen, 2001) have confirmed that stock prices globally are fractal.

In fact, their fractal properties exhibit a "fractal tree", which is nothing but a hierarchical model (HM) [55] (Sornette & Johansen, 1998). Furthermore, these fractal properties usually have sub-trees, with various other scaling properties. Coming to the Log Periodic Power Law (LPPL), it is a micro-state hierarchical fractal construct. Despite being generated from a micro-state, the LPPL usually offers a detailed macro-state overview. It has been observed that strong positive herding, based on a complex agent-based interaction (positive feedback) propels the asset class to reach extreme levels of valuation in almost no time, simply to face a second-order phase transition (crash) all of a sudden. Hence, both phases (growth and decay) are individually persistent, exhibiting long memory traits. Being fundamentally fractal, both phases of an underlying asset's bubble are self-similar or self-affine. Research proved that the build-up of the bubble, the critical point and finally the crash are completely based on a fractal premise. The very first representation of LPPL [49] (Johansen et al., 2000) was conceived with the same premise. Johansen et al. (2000) [49] had assumptions such as positive feedback and trader's affinity (group behaviour, with only buy and sell). It is important to note that, in the short run, traders do not hold onto their assets. Hence, it is believed that the LPPL takes into consideration positive feedback-based herding (generating speculative bubbles) following the inevitable crash. Financial mar-

kets are way too complex to follow such a straightforward rationale. Therefore the LPPL produces false alarms at times (Johansen, et al., 2000) [49]. For this reason, Johansen et al. (2000) [49] was recalibrated in order to develop a stronger model for future observations. Filimonov and Sornette (2013) [28] furthered the LPPL modelling and recalibrated it with less non-linear parameters, thus making the model robust [28] (Filimonov & Sornette, 2013).

We have investigated 11 stocks with the highest market capitalization (Orsted (DOGEF—Danish Wind Power company), Plug Power (PLUG—Hydrogen Fuel company), Next Era Energy (NEE), First Solar (FSLR), Enel (ESOCF), Iberdrola (IBDSF), Innergex (INGXF—Canadian Sustainable energy company), Solar Window Technology Inc (WNDW), Boralex (BLX), Azure Power (AZRE) and Canadian Solar (CSIQ)), across North America and the European Union (EU). Furthermore, we investigated the S&P Global Clean energy index, which indicates the global participation in this bubble, as it also includes developing nations. Moreover, the biggest Exchange Traded Fund (Invesco's solar ETF (TAN)) by market capitalization was also considered. Thus, we have examined 11 energy stocks, an ETF (Invesco's solar ETF) and an Energy Index (S&P Global Clean Energy Index) from the LPPL standpoint.

Our observations ranged from 31 December 2019 to 11 October 2021, where daily closing prices were taken into consideration. The especially reformulated LPPL model by Filimonov and Sornette (2013) [28] is quite efficient in finding bubbles in various asset classes. We have selected these 11 companies and an ETF based upon their origin (North America and the EU) and their listing (NASDAQ). We also selected the S&P Global Clean Energy Index to check the holistic impact on the entire sector. All data were acquired from Bloomberg and Thomson Reuters Datastream.

This study has been carried out by the recalibrated Filimonov & Sornette (2013) [28]. However, it all started with the Johansen–Ledoit–Sornette (2000) [49] model, having more non-linear parameters:

$$y_t = A + B\,(t_c - t)^\beta + C(t_c - t)^\beta \cos(\omega log(t_c - t)) + \phi \tag{1}$$

where $t_c$ denotes the most plausible time of the market crash, $\beta$ represents the exponent of exponential growth during both bubble and crash phases, $y_t$ is the expected value of the logarithm of price ($y_t > 0$), $\omega$ represents the angular magnitude of the oscillation during the bubble formation phase and $t$ is any time into the bubble preceding ($t < t_c$). $A$, $B$, $C$ and $\Phi$ are units having a less structural information, albeit they are coefficients. $A$, with a condition $A > 0$, signifies a bias term which can be ignored when prices are normalised. Moreover, $A$ is the price at the peak of the bubble. $B$ signifies the height of the bubble just before the inevitable crash ($B < 0$). $C$ is the magnitude of the oscillations around the exponential growth ($|C| < 1$). $\Phi$ is the phase shift parameter, from bubble to crash.

Fast movements (along with an angular frequency) generate the bubble from nowhere. It is propelled by the chain of positive feedbacks and reaches the critical point ($t_c$). The initial model of Johansen et al. (2000) [49] had too many non-linear estimates, making the outcome reasonably unstable. Furthermore, the Johansen et al. (2000) [49] model is difficult to optimise because of too many local minima. Even the Johansen et al. (2000) [49] model was not in harmony with back-propagation at that state. Hence, a logical recalibration was essential.

Therefore, Filimonov and Sornette (2013) [28] amended the Johansen et al. (2000) [49] construct model with less non-linear parameters:

$$yt = A + B(t_c - t)\beta + C_1(t_c - t)\beta \cos(\omega log(t_c - t)) + C_2(t_c - t)\beta \sin(\omega log(t_c - t)) \tag{2}$$

where $C_1 = C\,Cos\varnothing$ $C_2 = C\,Sin\varnothing$.

This recalibrated version of the LPPL algorithm has namely four linear variables ($A$, $B$, $C_1$, $C_2$) and three non-linear variables ($t_c$, $\omega$, $\beta$). These four linear parameters ($A$, $B$, $C_1$, $C_2$) are based on the "Standard slaving principle", with multiple self-organised subsystems (or microstates) constructing an entire system or macrostate. Haken introduced the slaving principle in order to understand a complex macrosystem or macrostate as an assembly

of many tiny non-linear microstates [56] (Haken, 1975). It has been a stylized fact for quite some time now that asset bubbles are nothing but a combination of self-organised, non-linear microstates with time-varying information. Therefore, the slaving principle suits perfectly. The subordination procedure has been used to propagate non-linear parameters ($\omega$, $\beta$) and to obtain regularity results. The Nelder–Mead simplex model has been used to find the local minima in a multidimensional space through the Filimonov and Sornette, (2013) [28] model.

Despite recalibration by Filimonov and Sornette, (2013) [28], the model remained sensitive to the input values. Usually, the bubble indicator is not consistent enough with any length of window. However, windows of observation shall be of optimum length. The range is quite wide, from a minimum of five trading days to a maximum of 750. The reformulated LPPL works well inside these conditions and, therefore, it will not suit the day traders, unlike the investors.

## 4. Empirical Results

Table 1 presents the conditions of the LPPL parameters, from a literature review. We have to check whether or not all 35 crashes in the Green Bubble fit in at the same set of values declared in Table 1. Models built showcased for $\beta = 0.33 - 0.18$, $\omega = 6.36 - 1.56$ and $\varphi = 0$ to $2\pi$. Drawdowns have been calculated using Sornette's method, the "price coarse graining" algorithm with $\varepsilon = 0$. Drawdown is the cumulative loss from one local maximum to the immediate next minimum; a size that is above the threshold '$\varepsilon$'.

**Table 1.** Stylized facts of LPPL.

| Parameter | Constraint | Literature |
|:---:|:---:|:---:|
| $A$ | ($>0$) | Kuropka and Korzeniowski, (2013) [57] |
| $B$ | ($<0$) | Lin, Ren, and Sornette (2014) [58] |
| $C_1$ | (Cos function) | Filimonov and Sornette, (2013) [28] |
| $C_2$ | (Sine function) | Filimonov and Sornette, (2013) [28] |
| $t_c$ | ($t$ to $\infty$) | Kuropka and Korzeniowski, (2013) [57] |
| $\beta$ | (0.1 to 0.9) | Lin, Ren, and Sornette (2014) [58] |
| $\omega$ | (4.8 to 13) | Johansen, (2003) [59] |

Note: The table above p exhibits the conditions of the LPPL parameters of Equation (2) used for empirical analysis.

Table 2 presents the coefficients of the LPPL parameters in Equation (2). Drawdown (DD) is the break between the local minima to the next local maxima and it is $\geq 17\%$. Table 3 represents the identified events behind eight bubble crashes ($>150\%$). A prominent LPPL signature is exhibited by all 35 bubble crashes across all 12 Green Energy companies and the S&P Global Clean Energy Index from 31 December 2019 to 11 October 2021 (overlapping with the COVID-19 period). We did not find any false positive alarm. Substantially lower levels of Root Mean Square Errors (RMSE) in all 35 cases support the LPPL fitment.

**Table 2.** Coefficients of the LPPL parameters having a drawdown of $>18\%$.

| Corporate/Index | Bubble | Time | $t_c$ | $A$ | $B$ | $C_1$ | $C_2$ | $\beta$ | $\omega$ | DD (%) |
|:---:|:---:|:---:|:---:|:---:|:---:|:---:|:---:|:---:|:---:|:---:|
| | B1 | 6 January 2020 to 4 March 2020 | 49 | 2.77 | −0.02 | 0.030 | 0.0019 | 0.49 | 7.75 | 31% |
| INGXF | B2 | 24 March 2020 to 20 October 2020 | 152 | 3.38 | −0.03 | 0.000 | 0.0015 | 0.63 | 10.42 | 86% |
| | B3 | 11 November 2020 to 14 January 2021 | 179 | 2.90 | 0.00 | 0.00 | 0.0048 | 0.70 | 7.72 | 42% |

**Table 2.** *Cont.*

| Corporate/ Index | Bubble | Time | $t_c$ | A | B | $C_1$ | $C_2$ | $\beta$ | $\omega$ | DD (%) |
|---|---|---|---|---|---|---|---|---|---|---|
| TAN | B1 | 23 March 2020 to 16 November 2020 | 178 | 1.47 | 117,695 | 602 | 3003 | 0.25 | 10.13 | 251% |
| | B2 | 17 November 2020 to 29 December 2020 | 29 | 4.79 | −0.05 | 0.006 | 0.000 | 0.63 | 7.61 | 51% |
| IBDSF | B1 | 9 January 2020 to 2 March 2020 | 44 | 2.56 | 0.00 | 0.000 | 0.0000 | 0.77 | 8.92 | 32% |
| | B2 | 11 May 2020 to 30 July 2020 | 58 | 2.59 | 0.00 | 0.001 | 0.000 | 0.71 | 12.31 | 42% |
| | B3 | 3 March 2021 to 20 May 2021 | 60 | 2.64 | 0.00 | 0.000 | 0.0000 | 0.43 | 12.16 | 18% |
| ESOCF | B1 | 15 May 2020 to 20 July 2020 | 53 | 2.16 | 0.00 | 0.000 | 0.000 | 0.18 | 11.56 | 57% |
| | B2 | 30 October 2020 to 20 January 2021 | 56 | 2.33 | 0.00 | 0.000 | 0.0001 | 0.32 | 7.18 | 39% |
| | B3 | 3 March 2021 to 26 April 2021 | 39 | 1.04 | 0.998 | 0.000 | 0.0001 | 0.134 | 12.66 | 19% |
| WNDW | B1 | 4 November 2020 to 8 January 2021 | 78 | 15.66 | 21.65 | −0.08 | −0.171 | 0.35 | 12.55 | 929% |
| | B2 | 4 May 2020 to 28 May 2020 | 21 | 0.07 | 0.64 | 0.55 | −0.46 | 0.59 | 12.06 | 134% |
| S&P CE | B1 | 3 March 2020 to 2 January 2021 | 13 | 9.60 | −0.30 | 0.00 | 0.00 | 0.70 | 10.10 | 183% |
| DOGEF | B1 | 31 December 2019 to 6 March 2020 | 161 | 4.70 | 0.00 | 0.00 | 0.00 | 0.30 | 9.60 | 36% |
| | B2 | 20 March 2020 to 27 July 2020 | 78 | 5.10 | 0.01 | 0.00 | 0.00 | 0.80 | 10.40 | 77% |
| | B3 | 23 September 2020 to 14 October 2020 | 18 | 7.10 | 1.20 | 0.00 | 0.00 | 0.70 | 11.00 | 23% |
| | B4 | 28 October 2020 to 1 December 2020 | 26 | 5.20 | 0.00 | 0.00 | 0.00 | 0.80 | 9.10 | 22% |
| | B5 | 7 December 2020 to 7 January 2021 | 19 | 5.50 | 0.00 | 0.00 | 0.00 | 0.50 | 6.90 | 32% |
| | B6 | 4 March 2021 to 9 April 2021 | 21 | 5.20 | 0.00 | 0.00 | 0.00 | 0.70 | 6.80 | 18% |
| | B7 | 23 June 2021 to 24 August 2021 | 47 | 5.50 | −0.04 | 0.00 | 0.00 | 0.80 | 11.70 | 22% |
| FSLR | B1 | 18 March 2020 to 28 August 2020 | 131 | 6.48 | −1.04 | −0.01 | −0.02 | 0.62 | 12.47 | 165% |
| | B2 | 28 September 2020 to 22 October 2020 | 28 | 5.32 | 0.03 | −0.01 | −0.03 | 0.56 | 7.42 | 39% |
| | B3 | 16 November 2020 to 21 January 2021 | 50 | 4.60 | 0.00 | 0.00 | 0.00 | 0.50 | 7.10 | 34% |
| | B4 | 8 March 2021 to 27 April 2021 | 41 | 4.30 | 0.10 | 0.10 | 0.00 | 0.71 | 12.43 | 25% |
| | B5 | 13 May 2021 to 12 July 21 | 47 | 4.00 | 2.50 | 0.20 | 0.10 | 0.71 | 8.80 | 35% |
| | B6 | 27 July 2021 to 13 September 21 | 34 | 4.70 | −0.10 | 0.00 | 0.00 | 0.83 | 11.60 | 30% |

**Table 2.** *Cont.*

| Corporate/ Index | Bubble | Time | $t_c$ | A | B | $C_1$ | $C_2$ | $\beta$ | $\omega$ | DD (%) |
|---|---|---|---|---|---|---|---|---|---|---|
| PLUG | B1 | 31 July 2020 to 9 October 2020 | 57 | 3.40 | −0.55 | −0.04 | 0.05 | 0.19 | 7.29 | 139% |
| | B2 | 6 November 2020 to 26 January 2021 | 54 | 14.51 | 20.19 | −0.09 | 0.05 | 0.11 | 7.64 | 288% |
| NEE | B1 | 2 January 2020 to 31 July 2020 | 182 | 4.20 | −0.01 | 0.00 | 0.00 | 0.53 | 8.15 | 22% |
| | B2 | 25 August 2020 to 28 January 2021 | 130 | 4.29 | 0.00 | 0.00 | 0.00 | 0.76 | 12.41 | 22% |
| | B3 | 18 March 2021 to 3 September 2021 | 138 | 4.46 | −0.29 | 0.29 | 0.00 | 0.75 | 9.90 | 18% |
| BLX | B1 | 19 March 2020 to 8 January 2021 | 214 | 4.16 | −0.10 | −0.01 | 0.00 | 0.53 | 11.40 | 203% |
| AZRE | B1 | 2 January 2020 to 14 January 2021 | 277 | 0.76 | 60.43 | 2.48 | 1.87 | 0.87 | 6.08 | 284% |
| CSIQ | B1 | 18 March 2020 to 21 January 2021 | 217 | 1.20 | 65.63 | −0.63 | 0.71 | 0.15 | 6.67 | 376% |

Note: The table above depicts the coefficients of the speculative bubbles in select Green Energy companies and the S&P Clean Energy Index from EU and North America, within the COVID-19 pandemic (31 December 2019 to 11 October 2021).

**Table 3.** Event linking with extremely large LPPL Drawdowns (> 150%).

| Sr. No. | Critical Date | Drawdown | Company/Index | Events |
|---|---|---|---|---|
| 1 | 8 Feburary 2021 | 929% | Solar Window Tech (WNDW) | 500% increase in prototyping and testing speed; 12-fold increase in testing capacity and output. |
| 2 | 21 January 2021 | 376% | Canadian Solar (CSIQ) | U.S. Energy forecasts showed crude oil production would fall from BPD 13.2 million in May 2020 to BPD 12.8 million in December 2020. |
| 3 | 26 January 2021 | 288% | Plug Power (PLUG) | U.S. Energy forecasts showed crude oil production would fall from BPD 13.2 million in May 2020 to BPD 12.8 million in December 2020. |
| 4 | 14 January 2021 | 284% | Azure Power (AZRE) | U.S. Energy forecasts showed crude oil production would fall from BPD 13.2 million in May 2020 to BPD 12.8 million in December 2020. |
| 5 | 16 November 2020 | 251% | Invesco Solar (TAN) | Brent came back to USD 43 a barrel after a long time. |
| 6 | 8 January 2021 | 203% | Boralex (BLX) | WTI Crude futures at USD 52 a barrel for third consecutive week. |
| 7 | 2 January 2021 | 183% | S&P Global Clean Energy | WTI Crude futures at USD 52 a barrel for third consecutive week. |
| 8 | 28 August 2020 | 165% | First Solar (FSLR) | Russia-OPEC Crude Oil Price war from March 2020 to July 2020 |

Note: This Table depicts eight bubble crashes (seven Green Energy companies and one Index) with a Drawdown more than 150%. Furthermore, it links those crashes with specific events (collected by the first Author from various credible sources).

All 35 past crash instances occurred with the following four stylized facts:

1. $\beta = 0.52 \pm 0.38$;
2. $\omega = 9.65 \pm 3.39$;
3. Minimum Drawdown (%) = 18%;
4. Average Drawdown (%) = 108%.

We have identified a total of 35 bubble crashes from 31 December 2019 to 11 October 2021 (Table 2). All of them matched the LPPL signature (Filimonov and Sornette, 2013) [28]. Interestingly, the minimum drawdown is significantly higher (18%) than that of most equities (7%) [30,31] (Ghosh et al., 2021; Ghosh et al., 2020) (Appendix A). It is a well-known fact that the average of the 10 worst equity drawdowns since 1970 until 2020 was 27% in the S&P-500 [60]. Studies across global indices during the mid-1980s to the early 1990s reveal some insightful information. It was found that the average drawdown (DD) was around 27% [50] (Johansen and Sornette, 2010), whereas the average drawdown (DD) of the Green Bubble is around 108% under stressful situations (e.g., COVID-19). This is four times more than the average drawdown. The S&P Clean energy index suffered from speculative bubbles, too, indicating the very fact that almost all of its elements have a bubble to varying degrees.

In fact, 40% of the Green Bubble has 50% or more draw-ups; 29% of the Green Bubble has 100% or more draw-ups just before its collapse (Table 2). As COVID-19 highlighted the importance of green energy, most European Union (EU) and North American Green Energy firms (or Clean-Tech firms) were sought out by investors, creating a kind of speculative bubble in a relatively short of time. Usually, a bubble is coupled with crashes, destroying investor's wealth and causing a dent in investors' confidence. Specifically, two of the bubbles (929% from WNDW (Solar Window Technology Inc., New York, NY, USA) and 376% from CSIQ (Canadian Solar)) are so steep that investors' wealth will most probably be destroyed.

However, an alternative viewpoint was suggested by Giorgis et al. (2021) [21] early in 2021. They documented that these bubbles are not strictly destructive in nature when looking through the lens of the Social Bubble Hypothesis. On the contrary, these bubbles would typically enhance economic activity and infrastructural spending. In fact, they weave a network of positive feedbacks in a repetitive manner among the high-net-worth individuals, venture capitalists and crucial government agencies in practically no time, resulting in an unprecedented increase in the growth of Clean-Tech or Green Energy companies. This alternative viewpoint opens up a whole new dimension of interpretations. Fundamentally, the influential social circles such as venture capitalists and high-net-worth investors understood the importance of alternate energy (Green Energy) more during the COVID-19 breakout. This realization translated into action when a substantial amount of new and even existing investments was after this handful of companies across the globe. This was the most likely reason of such a widespread bubble formation in this field. The Social Bubble Hypothesis (Giorgis, V., Huber, T., & Sornette, 2021) [21] indicated a sudden shift towards alternative energy forms as a possible substitute for the fossil-fuel-based "dirty energy". It is an investment opportunity for new entrants. Even players relying on crude oil have started their transition towards green energy, despite huge transaction costs. COP26 in Glasgow acted as a possible catalyst for such a sudden and fast-paced movement. This was definitely a build-up; however, the formation of the bubble in practically no time was inevitable.

More than 60% of our sample exhibited at least one Green Bubble with DD > 150% (Tables 2 and 3). Furthermore, we observed an intuitive association between the larger (DD > 150%) speculative bubbles with Crude Oil (Table 3). Despite the fall in Brent Oil, the New York Mercantile Exchange (NYMEX) and the West Texas Intermediate (WTI Crude futures) are enjoying positive correlations with the growth of these Green Bubbles (price surge of Green Energy companies). Interestingly, our findings are paralleled with an outstanding work of 2018, presenting the non-linear, co-integrating relationship between oil prices and Green Energy consumption [61–65] (Troster et al., 2018; Ma and Wang, 2022;

Frejowski et al., 2021; Vasylieva et al., 2021; Kaldellis, 2021). Perhaps this remarkable nexus is giving rise to FOMO (Fear Of Missing Out), a behaviour among investors of climate-focused companies [66]. Apparently, many traders have joined the Green Bubble bandwagon at its end, resulting in an unwanted position amid the global stock market rally in mid-2021. It is rather difficult for them to switch at the moment; however, at any given opportunity, they would switch. This would, in turn, build a bubble in the regular energy sector. Typically, there is a phase lag between the Green Bubbles and regular bubbles (formed in fossil-fuel-driven companies).

## 5. Conclusions

We have investigated Green Bubble behaviour in the stock prices of select stocks from a basket of Green Energy or Clean Tech companies (from North America and Europe), an ETF (Exchange Traded Fund) and the S&P Global Clean Energy Index during the COVID-19 pandemic. The covered period is from 31 December 2019 to 11 October 2021, where daily closing prices were taken into consideration. In this study, we adopt the log-periodic power law model (LPPL) methodology. Over the past decade, the LPPL model has been widely used for detecting bubbles and crashes in various markets (Brée and Joseph, 2013; Zhou, et al., 2018) [48,67].

Our analysis led us to the following conclusions: First of all, the presence of Green Bubbles in Green Energy companies during the COVID-19 is confirmed. Secondly, the average drawdown emerged as four times that of regular S&P-500 stock index under stressed conditions, such as COVID-19. Thirdly, it is understood that these bubbles will not typically destroy public wealth in the long run; on the contrary, they can increase the economic activity to a great extent, resulting in the sudden increase in the growth of Green Energy companies. Finally, the stylized facts obtained from empirical analysis would assist in predicting Green Bubbles in the future. Therefore, this study would most certainly assist policymakers, the industry and academia alike.

Given the growing importance of the role and contribution of stock exchanges to the challenges of global climate and economic developments, policymakers and the industry should remain committed to raise the awareness about the importance of green finance in securing a better tomorrow for future generations.

We can see Green Bubbles as a learning platform that help stock exchanges support the transition of green finance and take a leading role in creating more sustainable, creative and inclusive economies. Countries worldwide have to agree on the Sustainable Development Goals (SDGs), which include a clear call to action on climate change and economic development. We have identified 17 Sustainable Development Goals, set by the United Nations in 2015 for Sustainable Development [68]. Meeting these global goals will require a transition to green and sustainable, creative financial markets. There is a need for promotion of green energy products in particular, as well as of the ecological sector and the mainstream financial markets in general. For the time being, the Cap and Trade market is voluntary and the carbon footprint reduction with the purchase of rights is not actively encouraged. Their promotion would be good for reducing the global carbon footprint.

Green and sustainable finance offers benefits and opportunities for stock exchanges especially energy stocks. As a result, many businesses are focusing on sustainability and adopting an eco-friendly business model, which helps the environment, sustainability and attracts investors. So, there is a growing consensus about the role of stock exchanges and especially of energy stocks in promoting economic development and sustainability. Finally, there is a positive link between well-functioning financial markets—especially of green stocks and an economic and sustainable development. Clean-Tech firms promote environmental signals to future strategic partners, providing them with information on the green impact of their eco-innovations [69] (Rivas and Wigger, 2017). These positive signals create loops of positive feedback, resulting in the creation of the bubble.

**Author Contributions:** Conceptualization, B.G.; methodology, B.G.; software, B.G.; validation, B.G., and S.P.; formal analysis, B.G.; investigation, B.G. and D.K.; resources, B.G.; data curation, B.G.; writing—original draft preparation, B.G. and V.D.; writing—review and editing, S.P. and D.K.; visualization, B.G. and D.K.; supervision, B.G. and S.P.; project administration, B.G., S.P. and D.K. All authors have read and agreed to the published version of the manuscript.

**Funding:** This research received no external funding.

**Institutional Review Board Statement:** Not applicable.

**Informed Consent Statement:** Not applicable.

**Data Availability Statement:** Not applicable.

**Conflicts of Interest:** The authors declare no conflict of interest.

## Appendix A. Proof of LPPL Signatures

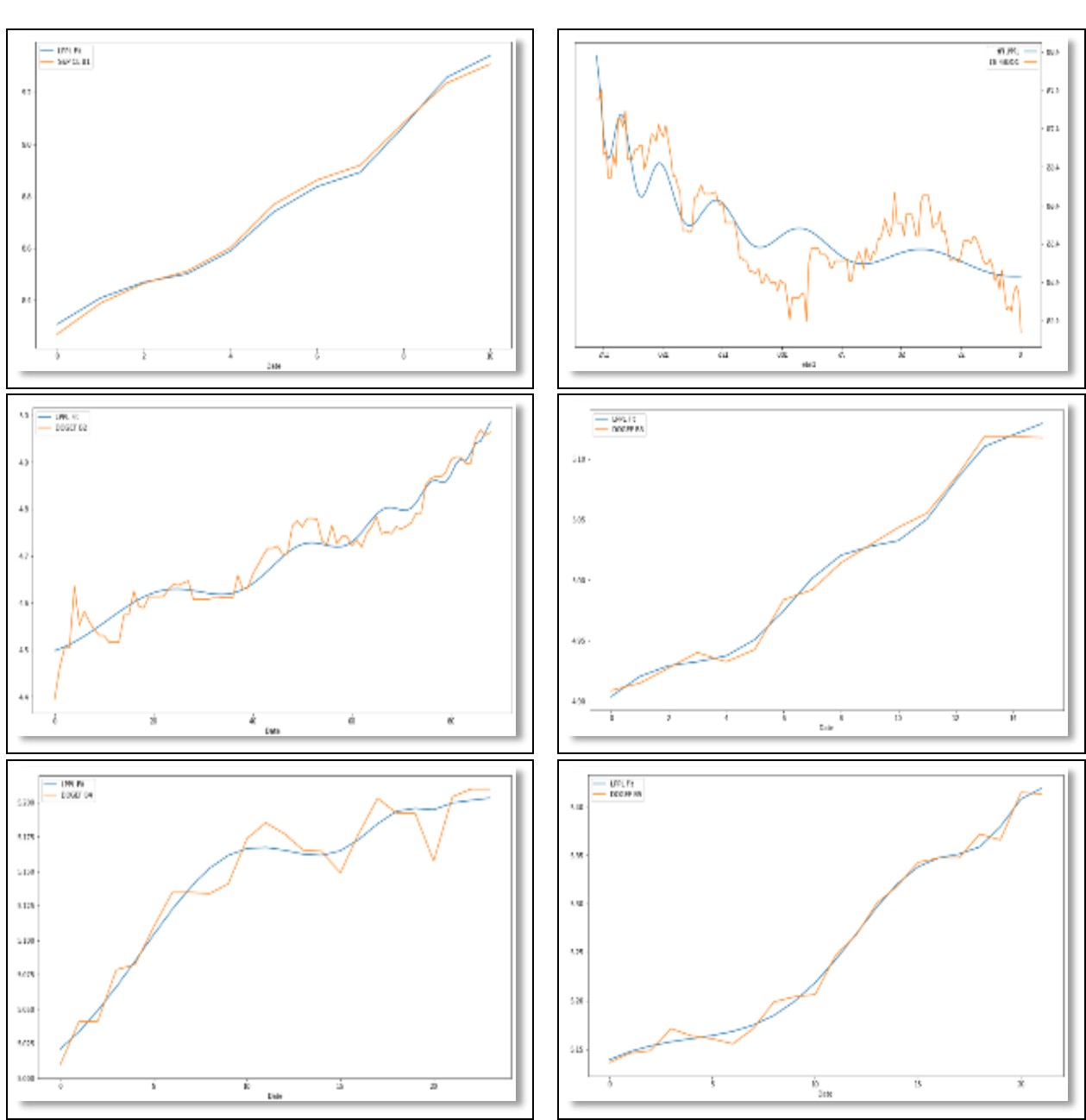

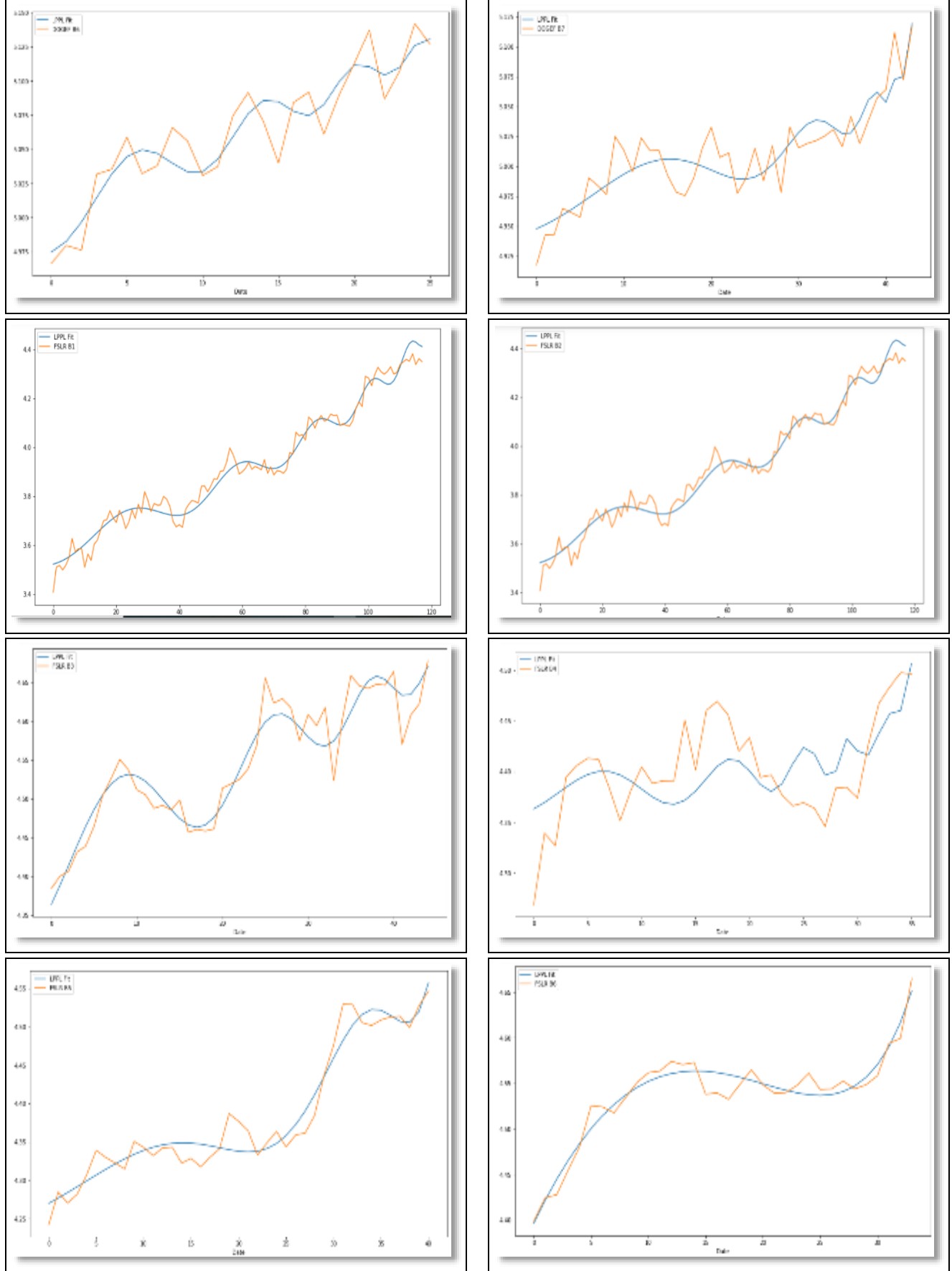

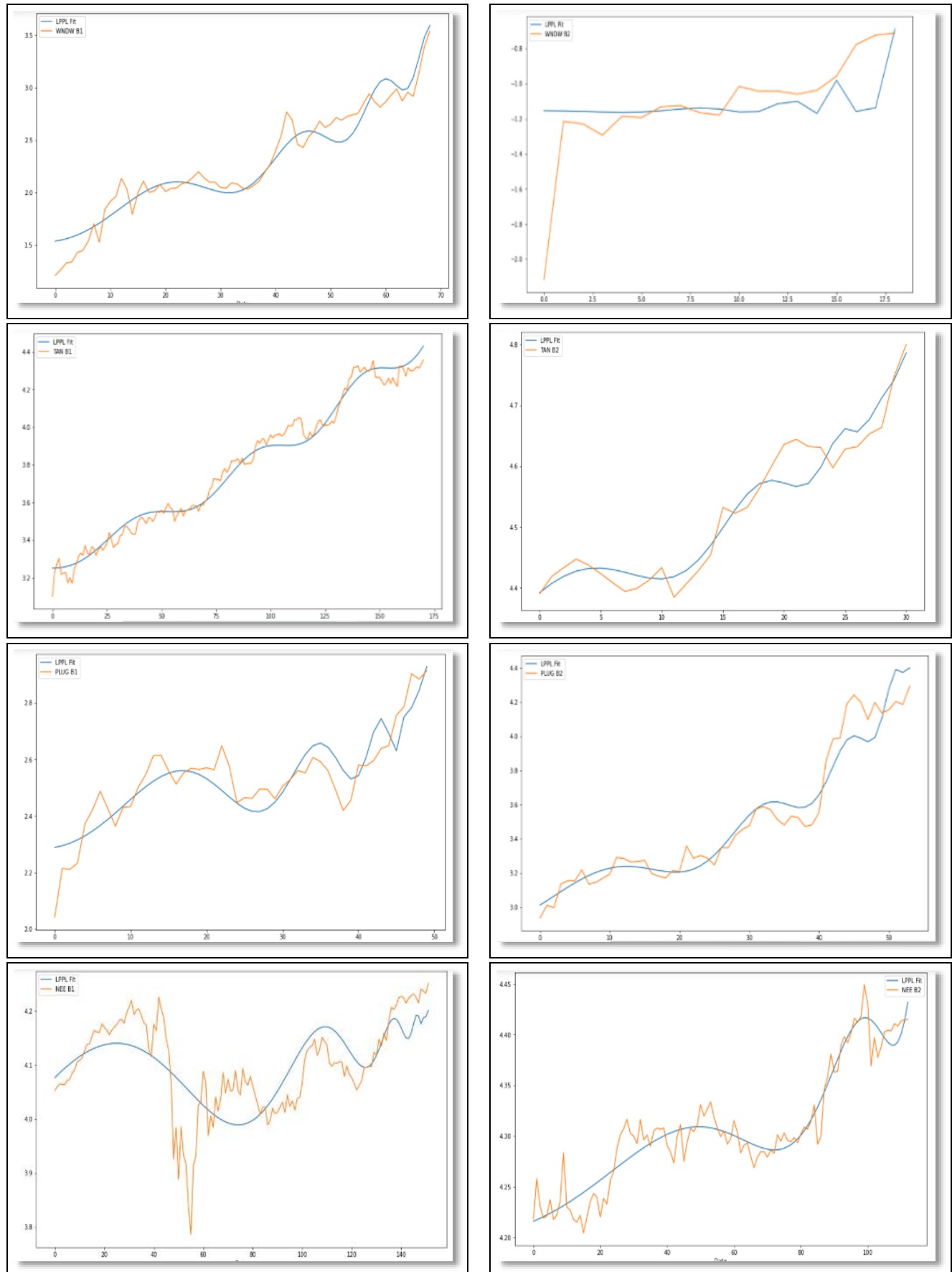

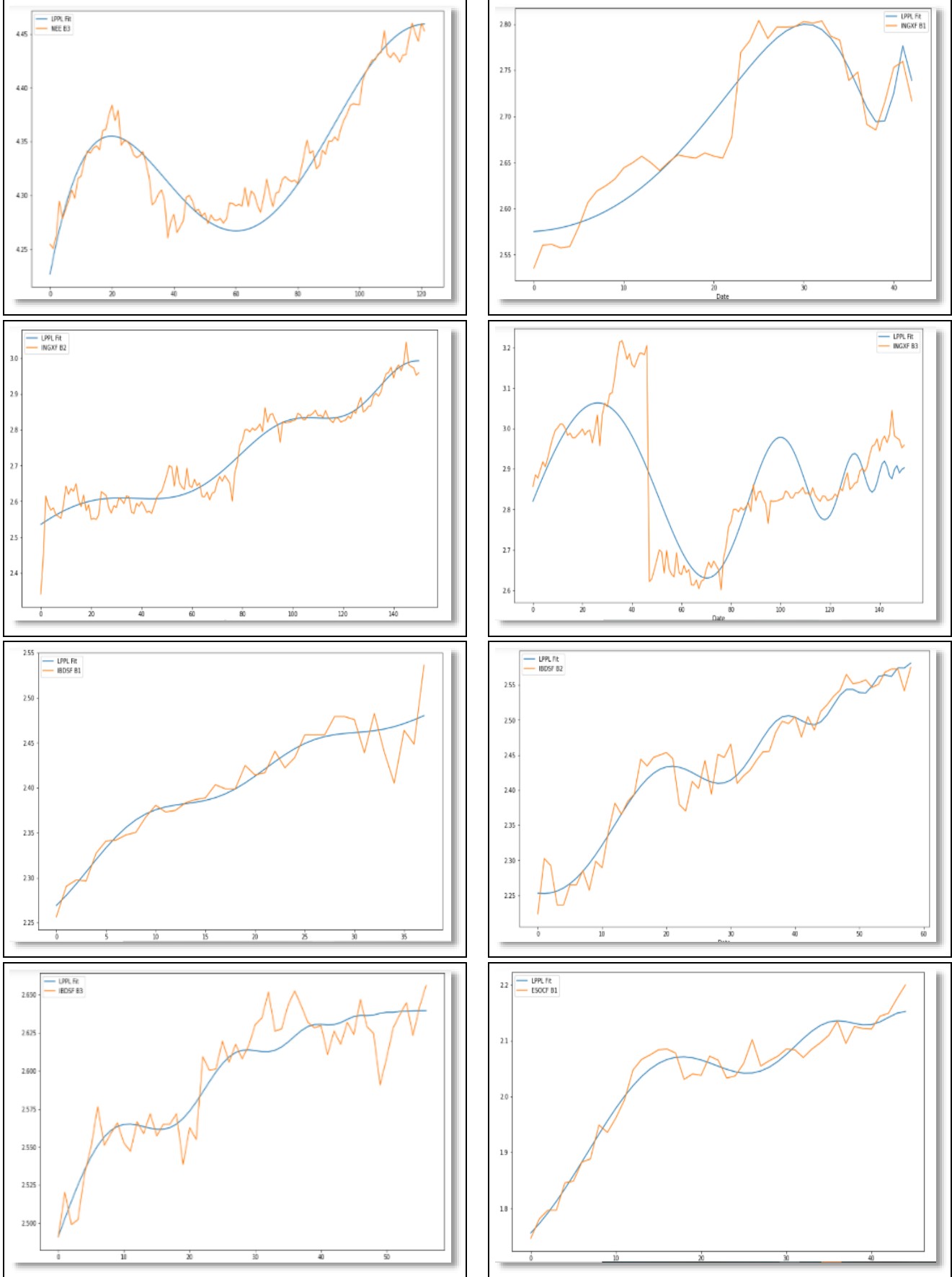

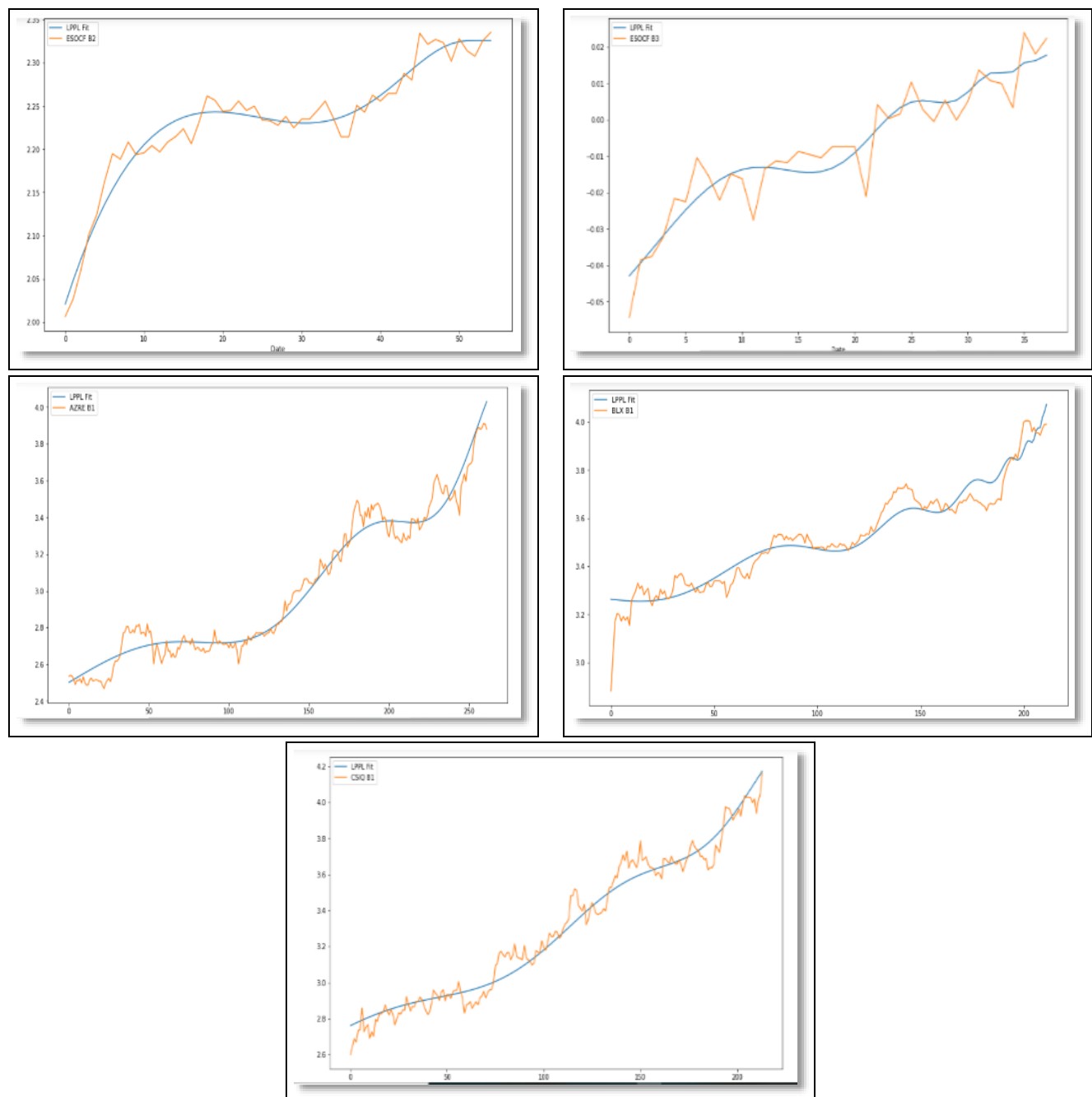

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
