# Peer review of "Deconstruction of the Green Bubble during COVID-19 International Evidence"

_sustainability, doi:10.3390/su14063466_

Round 1
Reviewer 1 Report
There is a great deal of merit in analyses that interact with the current pandemic context, especially in areas of opportunity such as the green movement. This is a wonderful topic.
Overall, this manuscript is fairly strong, but can be further strengthened by:
- Thorough citation of all premises such as lines 34-35; 35-36.
- Certain points should be expanded for specificity such as: “mandates” (L41); “SDGs” (L393-394); “need for promotion of green energy products” (L396-397).
- Concepts should be well-defined such as “green finance” (L42); “green energy” (L47); “Clean Tech Bubble” (L83); “Green Bubble” (L83-84); “Clean Energy” (L97); and Log Periodic Power Law (L217). It is good practice to define all concepts that are used, or relevant to the argument. You have done this with “Bubble” but similar treatment should also be given for the above concepts as well because they are typically confused and given false equivalence in the case of “clean” and “green”. Even if you do not ascribe to the stance of their uniqueness as concepts, it is still necessary to define them due to numerous other cases where they are treated as such. Noted, the above line references are the first instance of the term being used, it may be more sensible to place the definition at some other point where its definition is more relevant.
- Footnotes should contain relevant information. Please summarize the important parts of the information linked and cite it.
- Further explanation on the rationale for using LPPL.
- L155-156, if bubbles take some time to yield positive outcomes, it would be sensible to look at a brief history of green energy as well to fully understand the context and whether it is simply a bubble or a sustained movement and transition.
- L347-349, importance of alternate energy as what? A means of energy production? An investment opportunity? Was this a spontaneous realization? Was it a build-up?
Once again, I would like to reiterate: this article has many merits due to the specific context of the pandemic and subject of green technologies.
Reviewer 2 Report
The COVID-19 pandemic has impacted (and continues to do) the economy. It also has an impact on the development and investment in Green Energy. Authors investigated Green Bubble behavior in the stock prices of a selection of stocks, namely those with the highest market capital ization from a basket of North American and European Green energy or clean tech companies and the S&P Global Clean Energy Index during COVID-19 pandemic. The manuscript is interesting for the potential reader, but should be slightly improved:
- Websites should be added to the literature.
- Line 116-117 - the font is too large.
- Appendix Proof of LPPL Signatures is not readable.
- Please add more recent literature from "Sustainability" and "Energies".
- The chapter "Conclusions" should be expanded a little further.
- The English language should be checked by an English native speaker.
Reviewer 3 Report
The authors analyze the behaviour of the Green Bubble in the stock market. They emphasize that the emerging and a lot of promising green energy industry initiates the formation of Green Bubbles. The main conclusion of the article is that the Green Bubbles do not destroy public wealth in the long run, but on the contrary, they are capable to initiate the sudden growth of green energy companies.
The first part of the article "Deconstruction of the Green Bubble during COVID-19. International Evidence." reveals the main points of the topic. The second part of the article presents a literature review regarding the formation of the "bubbles". The third part of the article describes the research methodology and presentation of results. The last part of the article presents the conclusions.
